# Recycled Polyethylene Fibres for Structural Concrete

Jose A. Sainz-Aja [1], Marcos Sanchez [1], Laura Gonzalez [2], Pablo Tamayo [1], Gilberto Garcia del Angel [1], Ali Aghajanian [1], Soraya Diego [1] and Carlos Thomas [1,*]

1    LADICIM (Laboratory of Materials Science and Engineering), Universidad de Cantabria, E.T.S. de Ingenieros de Caminos, Canales y Puertos, Av./Los Castros 44, 39005 Santander, Spain; jose.sainz-aja@unican.es (J.A.S.-A.); marcos.sanchez@unican.es (M.S.); pablo.tamayo@unican.es (P.T.); gilberto.garcia@unican.es (G.G.d.A.); ali.aghajanian@unican.es (A.A.); diegos@unican.es (S.D.)
2    INGECID S.L. (Ingeniería de la Construcción, Investigación y Desarrollo de Proyectos), Av./Los Castros 44, 39005 Santander, Spain; laura.gonzalez@ingecid.es
*    Correspondence: thomasc@unican.es

**Featured Application: This research proposes a methodology and reports promising results concerning the valorisation of polyethylene waste as recycled fibres for use in the fibre reinforcement of concrete.**

**Abstract:** Modern society demands more sustainable and economical construction elements. One of the available options for manufacturing this type of element is the valorisation of end-of-life waste, such as, for example, the recycling of polymers used in industry. The valorisation of these wastes reduces costs and avoids the pollution generated by their landfill disposal. With the aim of helping to obtain this type of material, this work describes a methodology for recycling polyethylene for the manufacture of fibres that will later be used as reinforcement for structural concrete. These fibres are manufactured using an injection moulding machine. Subsequently, their physical and mechanical properties are measured and compared with those of the material before it is crushed and injected. The aim of this comparison is to evaluate the recycling process and analyse the reduction of the physical-mechanical properties of the recycled polyethylene in the process. Finally, to determine the properties of the fibre concrete, three types of concrete were produced: a control concrete, a reinforced concrete with 2 kg/m$^3$ of fibres, and a reinforced concrete with 4 kg/m$^3$ of fibres. The results show an enhancement of mechanical properties when the fibres are incorporated, particularly the tensile strength; and they also show excellent performance controlling cracking in concrete.

**Keywords:** recycled fibres; recycled polyethylene; fibre-reinforced concrete; sustainability

## 1. Introduction

The concrete industry is responsible for 10% of man-made $CO_2$ emissions [1,2] and for a large part of the volume of solid waste generated [3]. Being such a polluting industry, several authors have pursued different strategies to reduce the environmental impact of concrete. One of the most widely used methods is the replacement of natural aggregates with recycled aggregates from construction and demolition waste, resulting in what is known as recycled concrete. This is a well-studied topic, which has shown that it is possible to replace aggregates extracted from quarries with waste that would otherwise be sent to landfill, with slightly inferior mechanical and durability properties [4–13]. Another procedure studied to reduce concrete contamination is the use of self-compacting concretes, as they require a high volume of fines and are capable of absorbing a significant volume of waste in the form of fine particles [14–17]. Currently, due to the growing importance of fibre-reinforced concrete, the possibility of using recycled plastic in the concrete mix as a more environmentally friendly construction material is beginning to be considered [18–20].

Since the 20th century, the use of plastics has expanded in a wide range of products, supported by their good properties, such as low density, high strength-to-weight ratio,

high durability, ease of design and manufacture, and low cost [21]. The annual production of manufactured plastics has increased from 2 Mt in 1950 to 380 Mt in 2015, concluding that the total production amounts to 8.3 billion tonnes, of which only 9% has been recycled. As for thermoplastic polymers; historically, the most commonly produced would be: polypropylene (PP) with 21%, low-density polyethylene (LDPE) with 20%, high-density polyethylene (HDPE) with 16%, polyvinyl chloride (PVC) with 12%, and polyethylene terephthalate (PET) with 10% [22]. In Europe, the construction sector accounts for 20% of total consumption, the second largest sector after packaging [23]. One of the biggest problems is plastic that is inappropriately discarded or landfilled, as much of this waste ends up in the sea. Researchers Eriksen et al. estimated that there is almost 270 kt of plastic in the ocean [24]. A notable case of this problem is the "rubbish island" in the middle of the North Pacific Ocean. The characteristics of this island were also quantified, reporting an area almost three times the size of Spain [25]. The traditional linear model of plastic consumption based on: extract, make, use, and dispose is unsustainable. The best alternative for the plastics industry is the so-called circular economy, in which greater emphasis is placed on repairing or reusing products, thus extracting the maximum value from them during their useful life. At the end of their useful life, they are recovered and recycled [26]. The exponential growth of waste makes it essential to look for recycling alternatives, such as valorisation. In addition, there is growing concern and studies associated with the valorisation of different materials [27–30].

The use of natural fibres (e.g., straw or horsehair fibres) as reinforcement in construction has been widespread since historical times [31]. From the 1960s onwards, there was a quick development of fibre reinforcement in the construction sector, mainly motivated by the search for new structural materials with better mechanical properties. Fibres are composed of a wide variety of materials, including steel, glass and PP [32]. Nowadays, fibres have been shown to improve some properties of concrete, such as durability, elastic modulus, shrinkage control, and flexural strength, among others [33].

In particular, polymeric fibres are a suitable solution for structures exposed to moisture, as they are less prone to corrosion processes. In addition, they can reduce structural weight and control concrete cracking [34]. According to UNE-EN-14889-2 [35], polymeric fibres can be classified into two groups depending on their diameter: microfibres (less than 0.30 mm in diameter) or macrofibres. The latter are considered structural fibres.

The influence of polymeric recycled fibre on concrete properties have been studied in the last decade for several types of plastic, such as PET, PP, PVC or PE. A brief review of the mechanical properties of fibre-reinforced concrete has been carried out. Regarding compressive strength, different results can be observed; some researchers reported an enhancement of this property (PET [36,37], PP [38,39], PVC [40,41], PE [42–44]). However, other studies presented a clear reduction in strength (PET [45,46], PVC [47], PE [48,49]). Regarding flexural strength, all authors agree that it increases as fibres are added, this improvement occurs for a relatively low fibre content (approximately less than 1% fibre volume) (PET [36,37,45,46,50,51], PP [38,39,52,53], PVC [40,41,47], PE [42–44,48,49]).

In this context, this research involves the evaluation of out-of-use black polyethylene tubes for their recycling and subsequent manufacture of reinforcing fibres, which will be used for concrete. Section 2 explains the methodology (including the recycling process and the manufacture of the fibres and the concrete mix), together with a description of the experimental program. Section 3 reports the experimental results and the corresponding analyses, and Section 4 presents the main conclusions.

## 2. Materials and Methods

### 2.1. Recycling Process

The recycling process followed in this project consists of several steps: gathering material, cleaning, crushing, drying, and injection. The polymer selected for the work is polyethylene, specifically low-density polyethylene used in the industrial and construction sectors. In this sense, the first measure is the collection of the thermoplastic. For this

purpose, black pipes with blue bands with a diameter of 25 mm were collected, as shown in Figure 1a from Valoria S.L. Cleaning is a fundamental step, since impurities of any kind can cause problems in the cylinder and in the nozzle of the injection machine, as well as introduce heterogeneities into the final product, deteriorating its mechanical performance. For this purpose, the pipes were cut into smaller sections with an electric cutter and cleaned manually with water and a compressed air gun. The grinding of the pieces was carried out in a thermoplastic material recovery machine: the Matéu and Solé S.A. brand, model 19/25 M5-5; soundproofed with hardened steel blades, and equipped with sieves that reduce the size of the granules to 5–6 mm, as shown in Figure 1b.

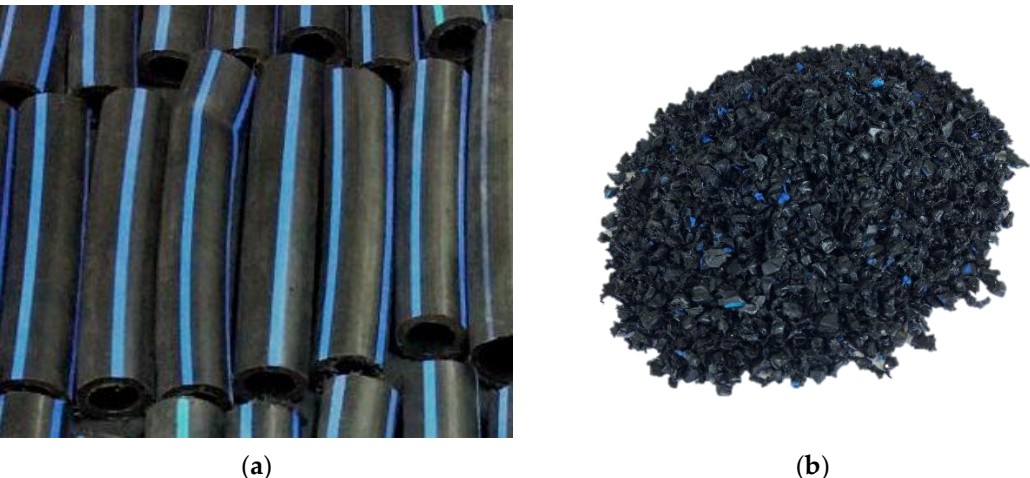

(**a**)   (**b**)

**Figure 1.** (**a**) Polyethylene pipes to be recycled. (**b**) Polyethylene pellets obtained from the crushing of the original material.

Before injection, the material is dried in an oven to avoid problems such as cavitation in the machine cylinder or steam bubbles generating pores in the final product. As polyethylene is a non-hygroscopic polymer, its capacity to absorb humidity is low, so a drying time of 2 h at 60 °C is enough.

The injection stage used the Arburg Allrounder model 221K injection moulding machine with 35 tonnes of clamping force and 49 cm$^3$ injection capacity, as well as a two-cavity A-type ISO mould with a Z-shaped feed channel, both shown in Figure 2. This machine was used to manufacture both the recycled polyethylene fibres and the standardised test specimens in order to study the material.

In this section of the work, the original polyethylene, which from now on will be abbreviated as "PT", and the recycled polyethylene, which will be called "PR", will be characterised; with the aim of comparing both materials and analysing their performance.

### 2.2. Polyethylene Properties

EN ISO 1183-1:2019 [54] was used to determine the density of both PT and PR. To determine the tensile strength of the PR material, standardised test specimens were manufactured, which were tested according to EN ISO 527-1:2012 [55]. For the PT, specimens with a standardised geometry were die-cut and then tested in a similar way to those described in the previously defined standard. Finally, tensile tests were also carried out on the fibres. In this case, it was not possible to manufacture standardised specimens, so the EN ISO 527-1:2012 standard was adjusted as far as possible to test the fibres.

Finally, the penetration hardness in the polyethylene was determined using the type D durometer, see Figure 3a, as the standard EN ISO 868:2003 marks [56].

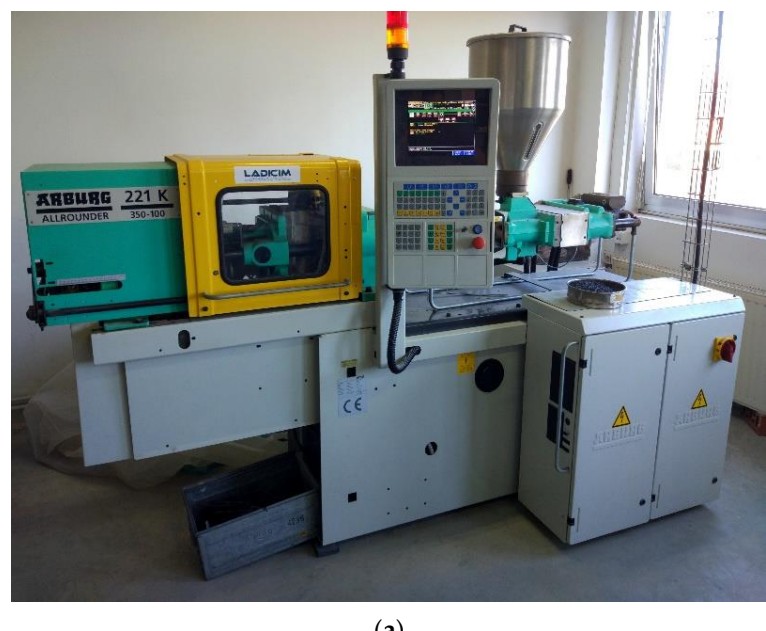
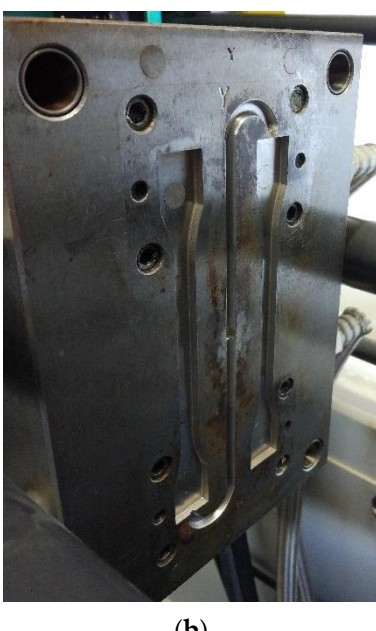

(**a**)  (**b**)

**Figure 2.** (**a**) Injection machine. (**b**) Mould used for standardised test specimens.

### 2.3. Fibre Manufacturing Process

The manufacturing process of the polyethylene fibres started from the recycled pellets that were previously characterised. The procedure carried out was as follows: the machine was switched on and the cylinder resistors were turned on to achieve the required injection temperature, in this case 230 °C. The pellets were then fed into the hopper (after the appropriate drying) and the cylinder was filled with the required quantity. The injection volume used here was 29 cm$^3$. As for the pressure introduced, after several tests it was concluded that the optimum pressure to achieve a fibre with a diameter of between 1–2 mm and which also acquired a certain roughness was with a pressure of 150 MPa. Once all the machine settings were established, the process began, and when the injected material came out, the fibre was pulled out manually, taking care not to reduce the section too much. Approximately 9 m from the machine, the bobbin (shown in Figure 3b) was placed, around which the wire was wound. Finally, a visual quality control was carried out on the fibre bobbins to rule out sections that did not have the appropriate section or whose surface was smooth, and the fibres were then cut manually to a length of 60 mm.

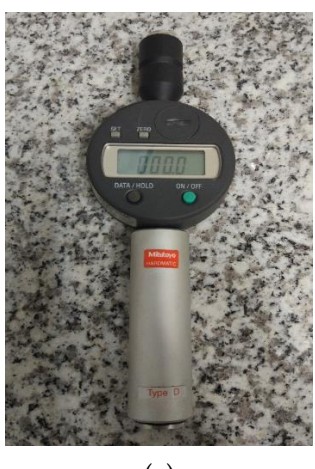
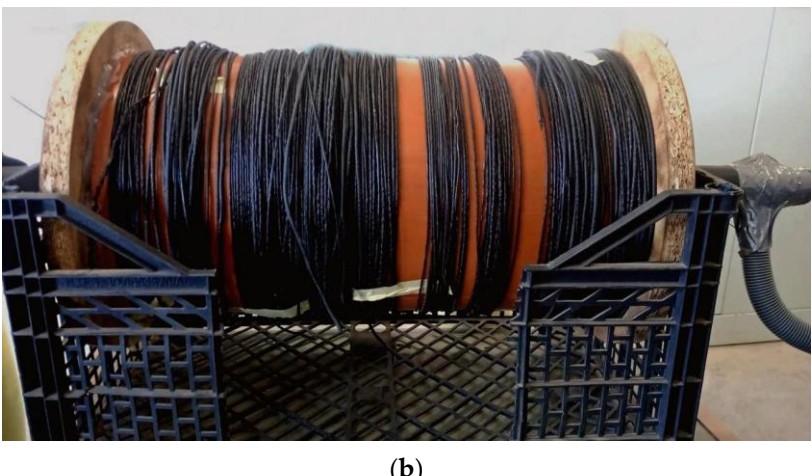

(**a**)  (**b**)

**Figure 3.** (**a**) Shore hardness meter type D. (**b**) Bobbin with valorised material.

### 2.4. Characterisation of Recycled Fibres

The polymer fibres used in concrete must fulfil both the mechanical and geometric performance requirements of the EN 14889-2:2008 standard [57]. A random sample of 100 fibres was used. Due to their surface irregularity, three diameter measurements were taken along the entire length of the fibre and the average was calculated. Moreover, a surface analysis of the fibre was carried out, as it plays a fundamental role in the adherence to the concrete matrix, calculating the roughness parameters in accordance with the EN ISO 4287:1999 standard [58]. A three-dimensional table TESA-Micro-Hite 3D was used to analyse the roughness of the fibres, as shown in Figure 4.

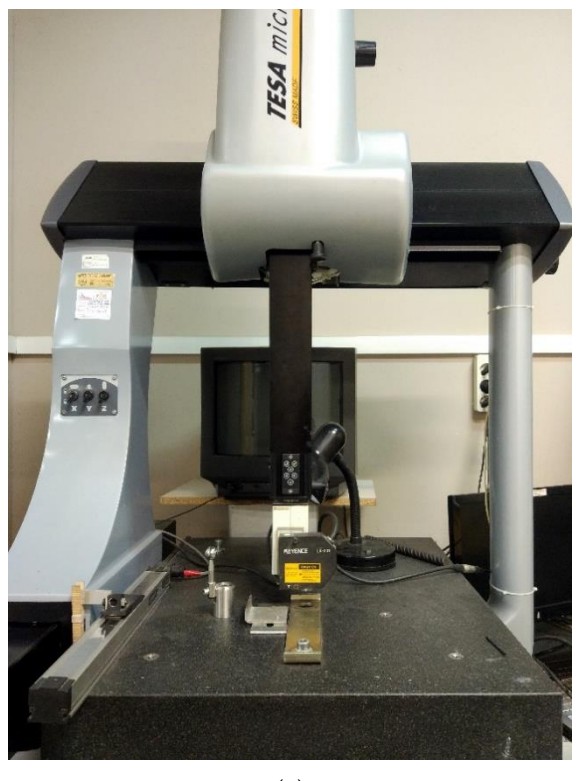
(**a**)

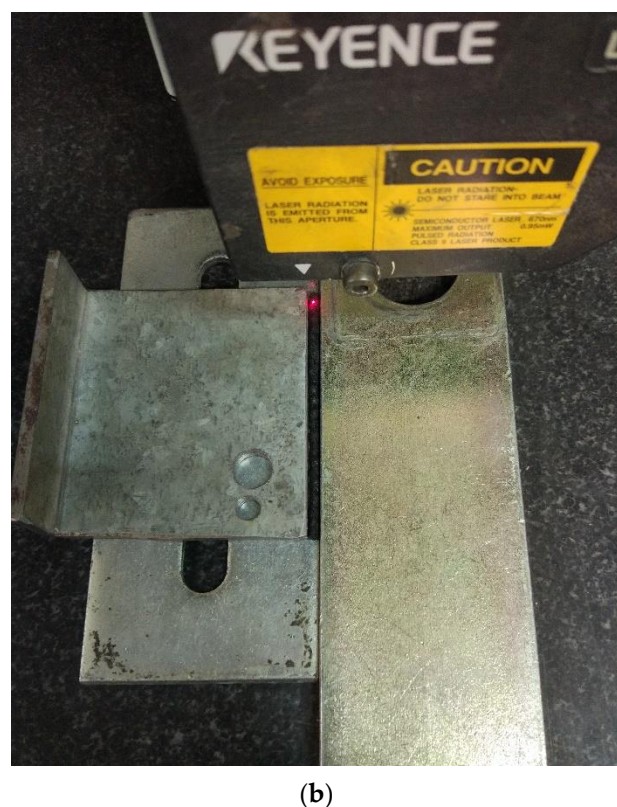
(**b**)

**Figure 4.** (**a**) Three-dimensional table. (**b**) Detailed fibre measurement.

### 2.5. Concrete Mix Proportions

A CEM III 32.5 SR type Portland cement was used for this work, in accordance with the EN 197-1 standard [59]. The concrete mixture was made with siliceous sand 0–2 mm, limestone sand 0–3 mm, and limestone gravel 6–12 mm. The grading curves of the aggregates are shown in Figure 5. A superplasticiser addition MasterEase 3850 was used to reduce the water and recycled polyethylene fibres. MasterEase 3850 is a highly active superplasticising and water-reducing admixture based on new polymer technology for the production of low viscosity concrete even at low water content.

A total of 3 different concretes were designed: a control concrete (HFC), a fibre-reinforced concrete with a volume of 4 kg/m$^3$ (HRFP-4), and finally, a fibre-reinforced concrete with a volume of 2 kg/m$^3$ (HRFP-2), see Table 1.

**Table 1.** Mix proportions.

| | Reference | High Fibre Content | Low Fibre Content |
|---|---|---|---|
| **Material (kg/m$^3$)** | **HFC** | **HRFP-4** | **HRFP-2** |
| Cement | 350 | 350 | 350 |
| Water | 210 | 210 | 210 |
| Sand 0/2 | 800 | 800 | 800 |
| Sand 0/3 | 623 | 623 | 623 |
| Coarse aggregate 6/12 | 600 | 600 | 600 |
| Superplasticiser | 5.3 | 5.3 | 5.3 |
| Polyethylene fibres | - | 4 | 2 |
| Water/Cement | 0.6 | 0.6 | 0.6 |

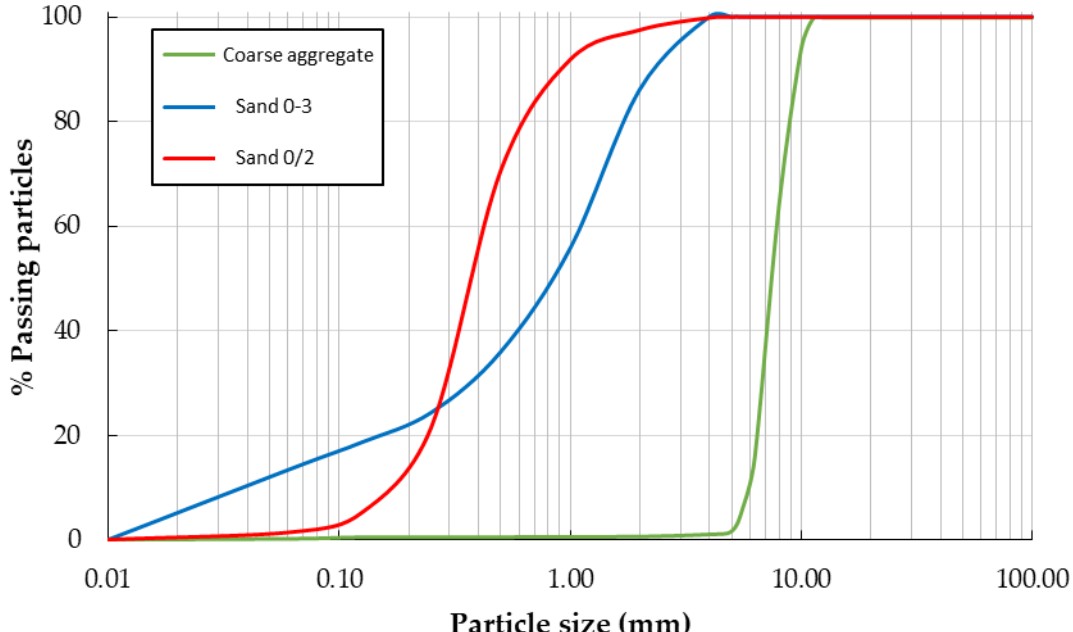

**Figure 5.** Grading curve.

### 2.6. Workability Tests

The Abrams cone test was carried out in accordance with the EN 12350-2:2009 standard [60], seen in Figure 6.

### 2.7. Physical Properties

To perform the physical characterisation tests on concrete, specimens obtained from cutting standardised cylinders (300 mm high and 150 mm in diameter) in 3 sub-samples of 100 mm high were used. The procedure followed to determine the physical properties of fibre-reinforced concrete, such as the density, porosity, and absorption coefficient, was based on the EN 12390-7:2009 standard [61].

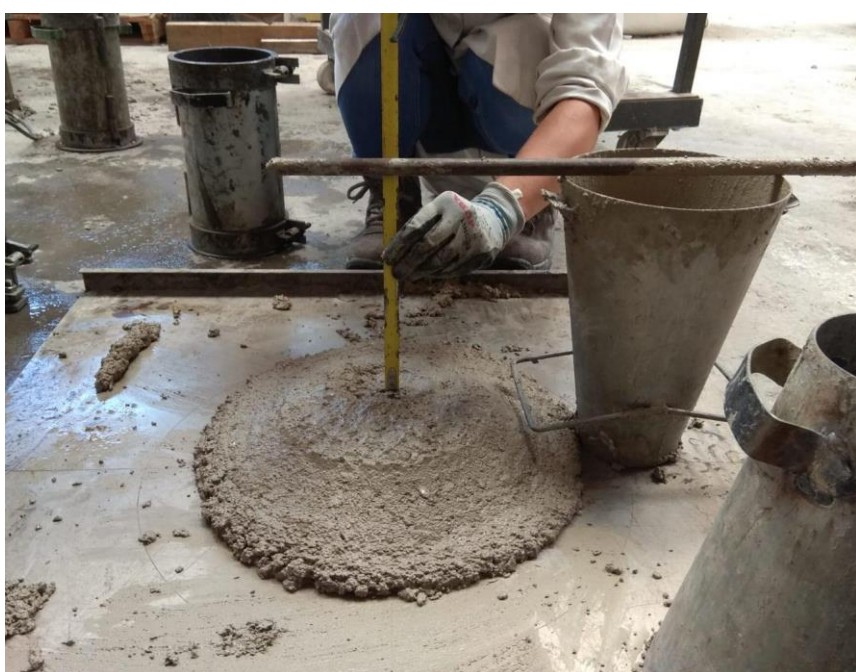

**Figure 6.** Measuring the height of the Abrams cone test.

### 2.8. Mechanical Properties

The determination of uniaxial compressive strength was based on the EN 12390-3:2009 standard [62]. In this test, standardised cubic specimens with 15 cm sides were used.

The tensile splitting strength tests were carried out in accordance with EN 12390-6:2009 [63]. This experiment was performed with samples cut from standardised cylindrical specimens; so, before proceeding with the test, a geometric characterisation of the specimen is required.

### 2.9. Durability

The oxygen permeability test was carried out in accordance with standards UNE 83966:2008 [64] and UNE 83981:2008 [65]. To conduct this test, samples previously used to determine the physical properties were taken.

The water permeability test was in accordance with EN 12390-8:2009 standard [66]. The same samples that were used to determine oxygen permeability were used for this test, see Figure 7.

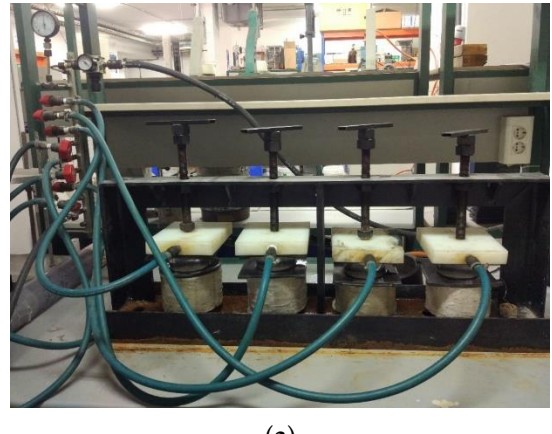

(**a**)

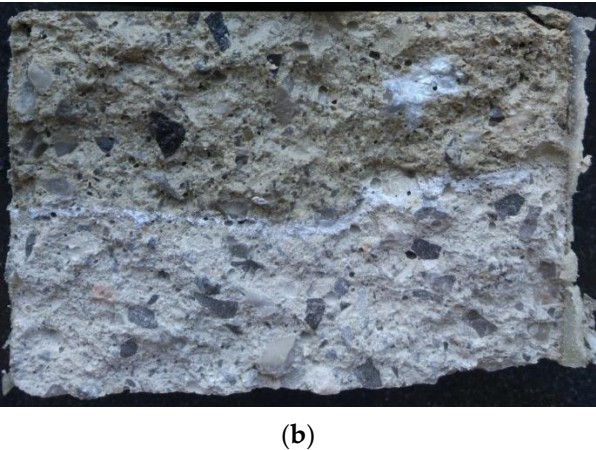

(**b**)

**Figure 7.** (**a**) Water permeability test layout. (**b**) Example of the moisture trace after the water permeability test.

### 3. Results and Discussion

*3.1. Characterisation of Polyethylene before and after Recycling*

3.1.1. Density

The mean density value of the polyethylene samples before recovery was 0.92 g/cm$^3$; in the case of valorised polyethylene the average value was also 0.92 g/cm$^3$. The constant density indicates that the recycling process did not affect this property. Furthermore, it can be seen that the results of the six samples were homogeneous. Therefore, it can be concluded that the material maintains its physical properties constant after recycling.

3.1.2. Mechanical Properties

Figure 8a shows the stress-strain curves obtained from the tensile test on the material before it was valorised. The elastic modulus of the material is calculated using the linear elastic section of the curve, for which a linear adjustment of the stress values between 0% and 5% strain was done, as shown in Figure 8c. Figure 8b,d are identical curves but they represent the behaviour of the material once it had been valorised.

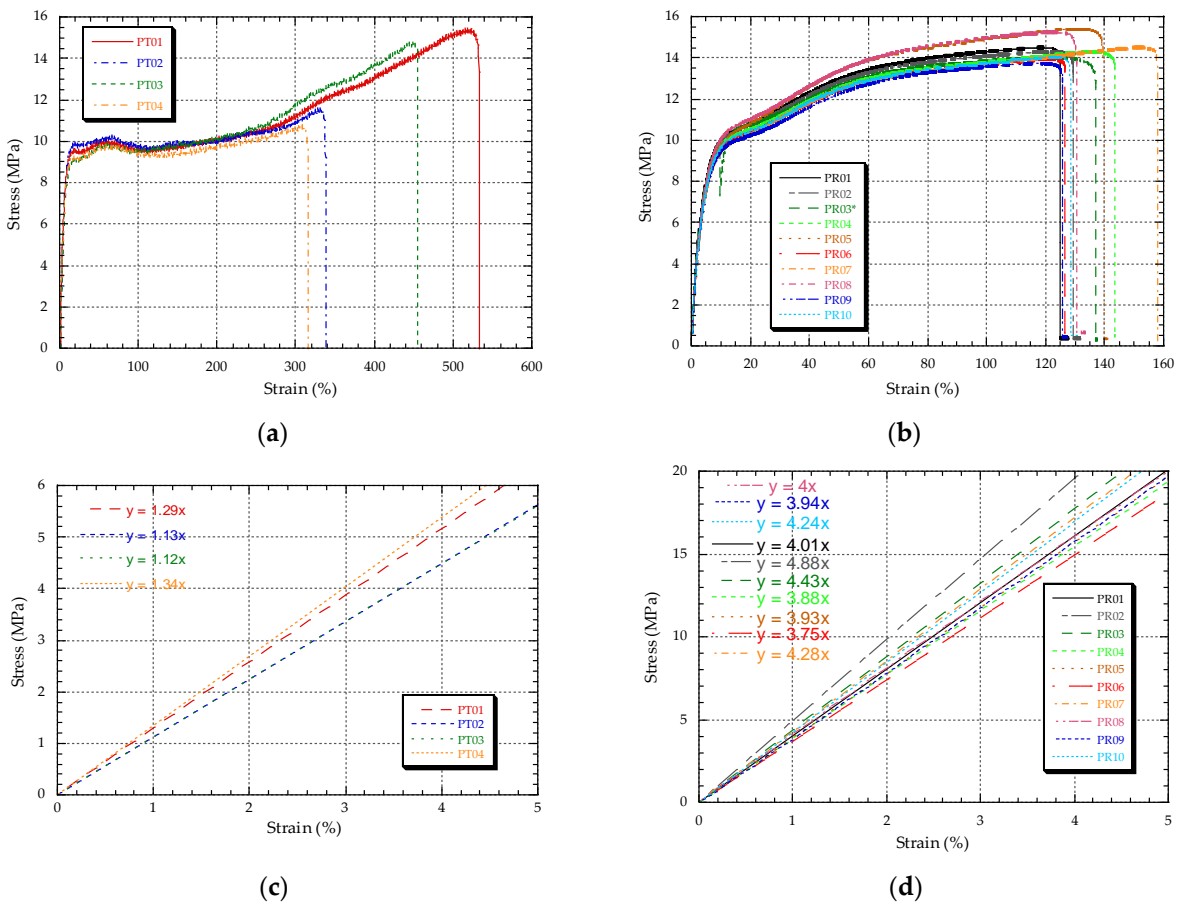

**Figure 8.** Stress-strain curves of polyethylene before and after being recycled. (**a**) Full stress-strain curves of material before it is valorised. (**b**) Full stress-strain curves of material after it is valorised. (**c**) Elastic section of the stress-strain curves of material before it is valorised. (**d**) Elastic section of the stress-strain curves of material after it is valorised.

Considering the stress-strain curve, it can be seen that the greatest increase in load was produced in the elastic section. Once the elastic limit had been exceeded, the material showed a yield plateau where, without increasing the load, the specimen reached deformations of around 200% (it was stretched to twice its initial length). Table 2 shows a summary of the main mechanical properties of polyethylene before and after recovery.

**Table 2.** Mechanical properties of polyethylene before and after being recycled.

|  | Elasticity Modulus (MPa) | Yield Strength (MPa) | Breaking Strength (MPa) | Strain under Max. Load (%) |
|---|---|---|---|---|
| Material before valorisation | 121.93 | 9.55 | 13.16 | 425.62 |
| Material after valorisation | 194.80 | 10.17 | 14.28 | 136.77 |

The material parameters were obtained before its valorisation. The mean value of the elastic modulus was 121.3 MPa, so it is a semi-rigid plastic ($E_t$ between 70 and 700 MPa). This indicates that in the elastic section, the increase in load produces relatively small deformations. At the breaking point there was an average stress value of 13.16 MPa and an average strain of 452.62%, see Figure 9. However, when analysing the results individually, it can be seen that, in the elastic section, the behaviour of the material remained almost constant, while in the plastic zone the results obtained were more heterogeneous, which means that the behaviour at rupture is more difficult to predict. On the other hand, the high percentage of strain indicates that it is a very ductile material. This characteristic is attributed to LDPE being a polymer with a branched chain structure. The ductility value also indicates that it is a polymer with high toughness, understanding this parameter as the amount of energy that a material absorbs before breaking.

In the case of the material already valorised, the mean values obtained were: an elastic modulus of 194.80 MPa, yield strength of 10.17 MPa, and a breaking stress of 14.28 MPa. As far as ductility is concerned, the strain under maximum load was 136.77%. The results obtained from the different specimens show a low dispersion, so that the behaviour of the recycled polyethylene is almost constant. This is an indication that in the recycling process there were no problems of homogenisation or high porosity that would cause drastic drops in the material's strength.

If the change in behaviour of the material after its valorisation is analysed, the first thing that can be observed is the values of both breaking stress and yield stress have increased by 6% and 9% respectively, so it can be concluded that the recycling process has not affected the strength of the material. However, the elastic modulus of the recycling was increased by approximately 59%. At the same time, the strain under maximum load of the material decreased by almost 70%; this also indicates that the tenacity of the recycled material is lower.

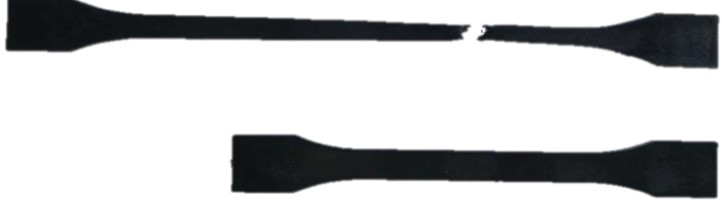

**Figure 9.** Example of the effect of the test on a specimen.

### 3.1.3. Hardness

Figure 10 shows a comparison of the Shore hardness D results on the material before and after assessment.

The average shore hardness of the original polyethylene was 47.54, being 46.9 for the recycled polyethylene; in terms of shore hardness D, these values indicate that this polymer can be considered hard. In view of the results, it can be concluded that the hardness of the material was not affected by recycling. On the other hand, although the hardness in polymers cannot be directly related to tensile strength (as it is in steels), it gives us an idea that the strength of recycled polyethylene must be similar to that of the original, as shown in previous sections.

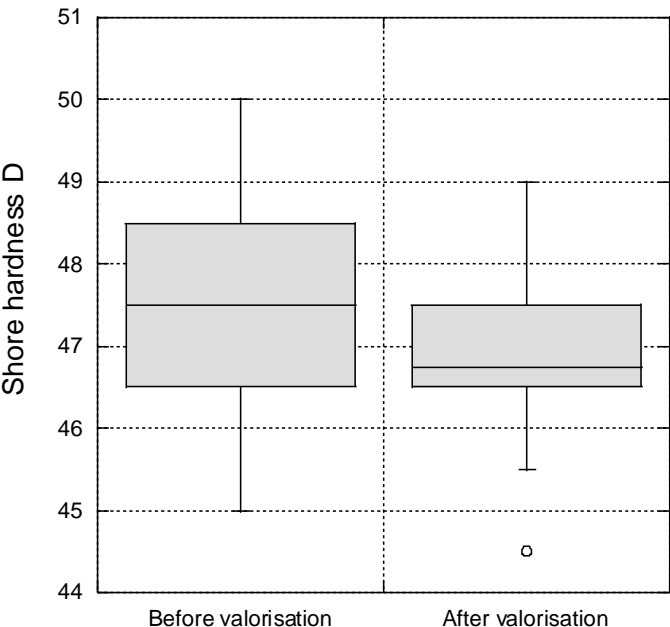

**Figure 10.** Shore hardness of polyethylene before and after being recycled.

### 3.2. Characterisation of Recycled Polyethylene Fibres

3.2.1. Geometric Properties of Recycled Fibres

For the analysis of the geometrical properties of the fibres, firstly, the diameter of 100 fibres was measured, obtaining an average value of 1.730 mm with a standard deviation of 0.365 mm. Therefore, it can be seen that the thickness of these fibres is considerably greater than the fibres usually used.

Subsequently, the average roughness of the fibres was analysed, see Figure 11a, obtaining a value of 0.1173 mm with its corresponding standard deviation of 0.0352 mm. In this case, obtaining fibres with a high roughness, as shown in Figure 11b, is beneficial for the application in concrete because it increases the friction between the two materials, which increases the adhesion of the fibres to the concrete matrix, and therefore, improves the ability of the fibres to reinforce the concrete.

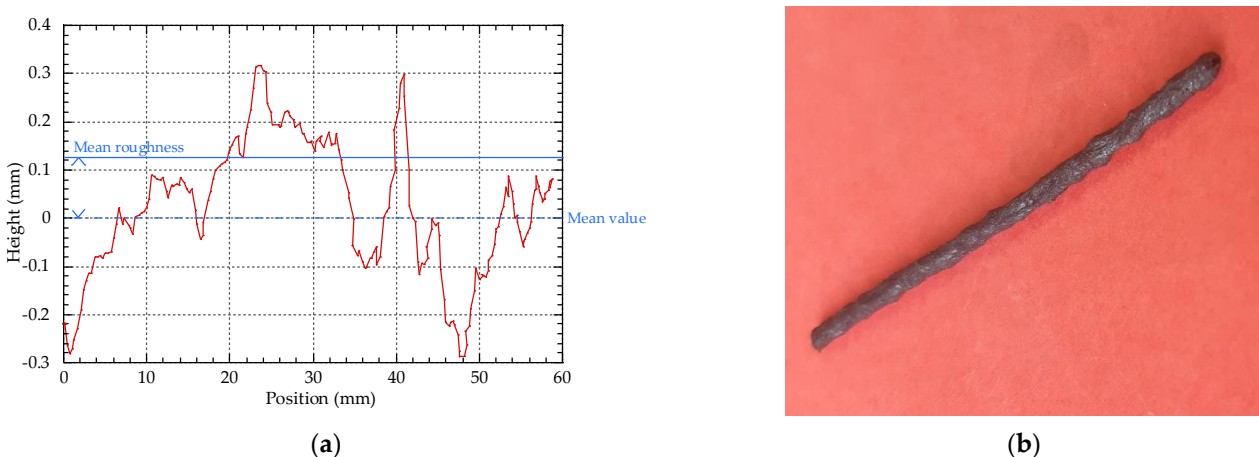

(**a**)  (**b**)

**Figure 11.** (**a**) Example of the determination of average roughness. (**b**) Example of recycled fibre.

As shown in Table 3, the fibres manufactured can be considered long, thick, and irregular. According to EN 14889-2 these fibres are classified as class II macro fibres and may have a structural function.

**Table 3.** Recycled fibre general properties.

|  | Mean Diameter (mm) | Mean Length (mm) | Density (g/cm³) | Mean Roughness (mm) |
|---|---|---|---|---|
| Recycled fibres | 1.73 | 60 | 0.92 | 0.1173 |

### 3.2.2. Fibre Tensile Strength

By analysing the stress-strain curve, see Figure 12, it can be observed that the behaviour of the material is very different between the elastic section and the plastic section. In the elastic zone, it is a relatively stiff material (i.e., to achieve deformation it is necessary to apply a high load). However, once the elastic limit is exceeded, the fibre behaves as a very ductile material; with low load increases, large deformations are achieved. There is also a lot of dispersion in the stress values; this is because the cross-section varies along the entire length of the fibre.

Regarding the results, a modulus of elasticity of 148.09 MPa was obtained, which represents an increase of 27% compared to the original polyethylene. The yield stress is 10.03 MPa, and compared with the original, this characteristic remains constant. The tensile strength obtained is 17.24 MPa, which constitutes an improvement of 31%. As for the deformation under maximum, it is 577.78% with respect to the original material, an increase of approximately 35%. This increase also indicates that the energy needed to break the fibre is greater, which means an increase in the toughness of the material.

Once all the results were analysed, it could be concluded that the fibres obtained improve their mechanical properties with respect to the original polyethylene. Therefore, the methodology followed in this work for the recycling and manufacture of the fibres is optimal for the application to which they are addressed.

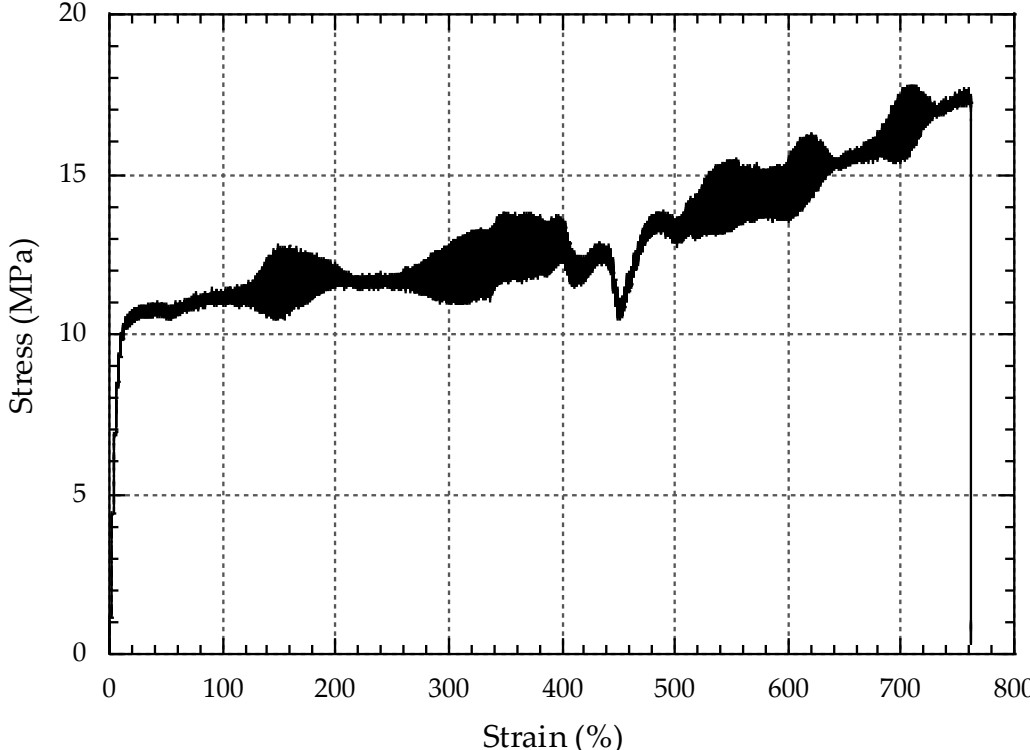

**Figure 12.** Example of a fibre stress-strain curve.

### 3.3. Comparison among the Properties of the Original Polyethylene, the Recycled One and of the Fibres

Figure 13 shows a comparison of the mechanical properties of polyethylene before being recycled, after being recycled, and the fibres made from recycled polyethylene.

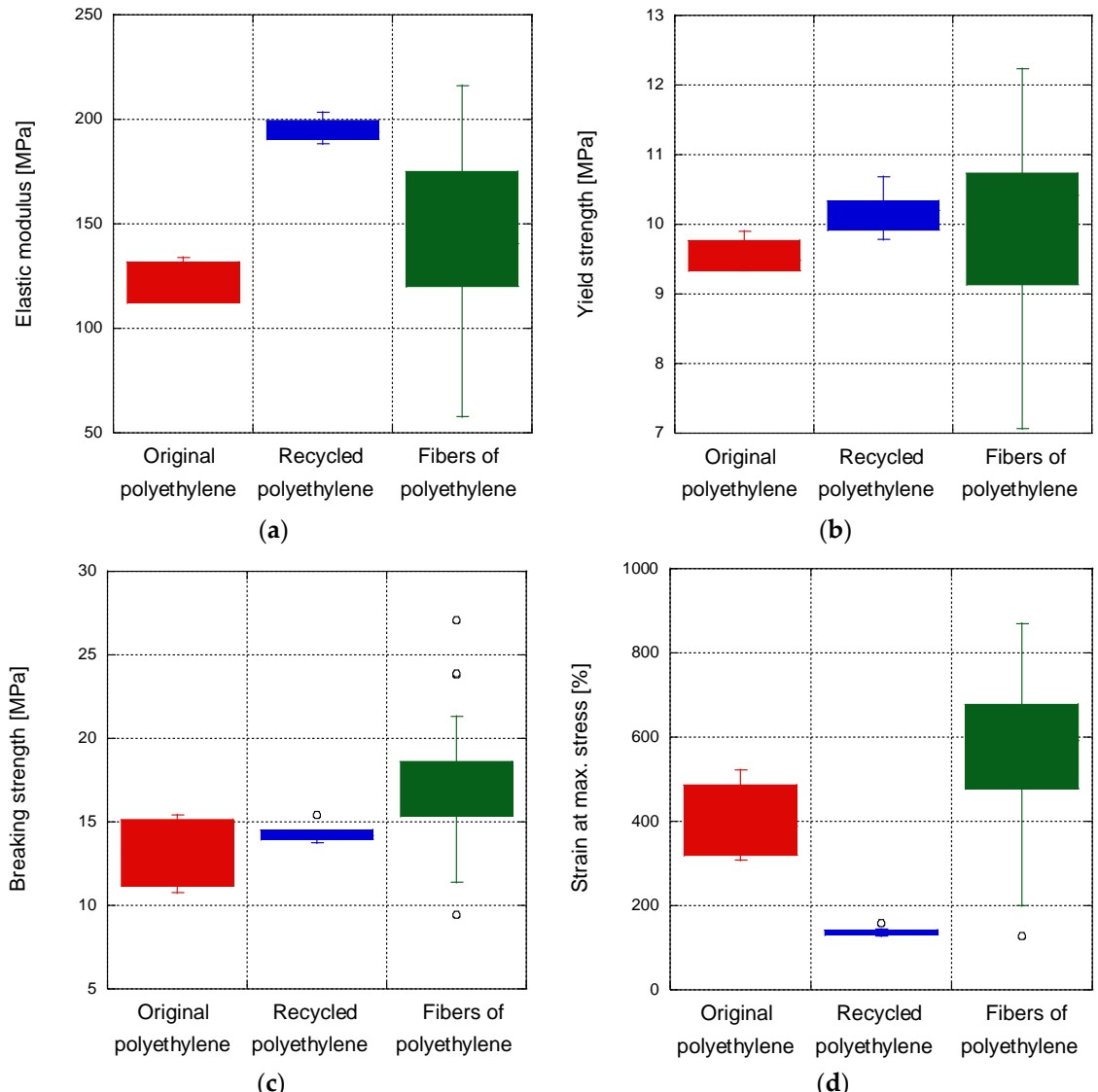

**Figure 13.** Comparison of the mechanical properties of polyethylene before and after valorisation and of polyethylene fibres. (**a**) Elastic modulus. (**b**) Yield strength. (**c**) Breaking strength. (**d**) Strain at maximum stress.

First of all, it can be observed from all of the graphs that the fibres are very scattered in relation to the other two products. This phenomenon is because, while the characterisation of the original polyethylene and the recycled one were carried out with standardised test specimens, the fibres are manufactured manually, which leads to irregularities in the geometry.

Figure 13a shows that the elastic modulus of both the PR and the fibres is increased in comparison to PT. Regarding the yield stress and breaking stress, although the mean values of both the valorised material and the fibres show a slight increase, the three products shown in Figure 13b,c indicate that the recycling process does not significantly modify these properties.

The main difference is the strain under maximum load, shown in Figure 13d. In the case of the fibres, although the average value is higher than that of the polyethylene before being valorised, the statistical distribution of the results indicates that the ductility of both is similar. However, the strain values of the recycled polyethylene are much lower than the other two. On the one hand, this reduction in ductility can be explained by the increased stiffness shown by the material. On the other hand, it could be due to aspect

ratio differences between the samples used; the higher aspect ratio they show, the lower the ductility they will have.

The differences in the manufacturing process between the fibres and the valorised polyethylene are in the final steps. In the case of the fibres, the polyethylene in its cast state leaves the cylinder nozzle at a pressure of 150 bars and is then pulled. This produces a stretching of the material so that the molecules are forced to move in one direction, causing a slight induced crystallisation of the polymer. In the case of the PR, the molten material is injected into a mould at a pressure of 1000 bar and a post-pressure is applied for a few seconds and then the mould is opened. Knowing that the pressure with which the PR is formed is almost eight times greater than that of the fibres and considering that the PR cools down more slowly than the fibres, it can be deduced that the PR is much more compact, which can explain why the PR is more rigid and less ductile than the fibres.

### 3.4. Concrete with Recycled Fibres

3.4.1. Concrete Workability

Table 4 shows the results obtained from the Abrams cone. From these results, it can be concluded that the presence of fibres in the studied range does not affect the concrete's workability.

**Table 4.** Concrete workability results.

| Mix | Cone (cm) |
|---|---|
| HFC | 24 |
| HRFP-2 | 23 |
| HRFP-4 | 23 |

Based on the results shown in Table 5, it can be seen that the presence of fibres barely changes the concrete workability. However, despite the limited influence of fibres on the fresh state behaviour, a slight reduction in the slump flow can be observed. This reduction in the workability of the concrete due to the presence of fibres agrees with the results obtained by other authors [67].

3.4.2. Physical Properties

Table 5 shows the physical property values of the manufactured concretes.

As far as the density of the concrete is concerned, it can be seen that there is no clear difference between control concrete and concrete with fibres. Furthermore, the values obtained are within the density range (2–2.6 g/cm$^3$) established for normal concrete according to EHE-8.

**Table 5.** Concrete physical properties.

| Mix | Bulk Specific Gravity (g/cm$^3$) | Apparent Specific Gravity (g/cm$^3$) | Absorption (%) | Porosity (%) |
|---|---|---|---|---|
| HFC | 2.20 | 2.33 | 2.52 | 5.55 |
| HRFP-2 | 2.18 | 2.28 | 2.00 | 4.35 |
| HRFP-4 | 2.26 | 2.36 | 1.91 | 4.31 |

However, the inclusion of 2 kg/m$^3$ and 4 kg/m$^3$ of fibres reduces the absorption by 21% and 24%, respectively. In the same way, the porosity of the concrete with fibres is reduced in both cases by 22% with respect to the control concrete. Figure 14 shows how the properties of porosity and absorption can be fitted to an exponential curve with respect to the volume of fibres contained in the concrete, so that the incorporation of fibres will

tend to reduce these properties, always considering that they are theoretical curves and that there will be a limit to the volume of fibres.

Connecting the porosity and the density of HRC and HRFP-4, it can be deduced that, just as the paste fills the gaps in the concrete, the fibres fill the gaps between the aggregates. Therefore, the reduction in porosity leads to a reduction in the number of gaps; this leads to a slight increase in the mass (concrete + fibres) for the same volume of specimen, which would explain the slight increase in the bulk specific gravity of HRFP-4. In the case of HRFP-2, although there is also a decrease in porosity, the apparent density is practically the same as that of HRC; this effect may be due to the fact that the polyethylene fibre has a lower density than that of the concrete.

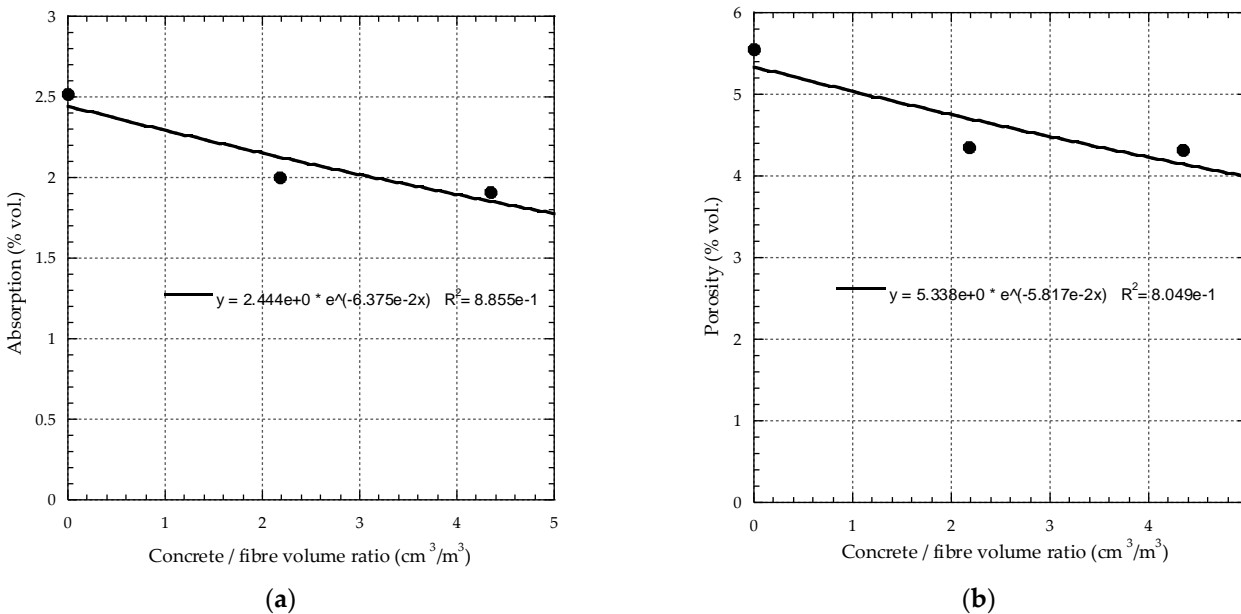

(a) (b)

**Figure 14.** Comparison of the absorption (**a**) and the porosity (**b**) of the concrete as a function of the concrete/fibre volume ratio.

### 3.4.3. Mechanical Properties

Table 6 shows the evolution of the mechanical properties of the three designed concretes.

**Table 6.** Concrete mechanical properties.

| Mix | Age (Days) | Compressive Strength (MPa) | Tensile Splitting Strength (MPa) |
|---|---|---|---|
| HFC | 7 | 26.01 | 2.02 |
| | 28 | 27.16 | 2.07 |
| HRFP-2 | 7 | 27.52 | 2.99 |
| | 28 | 28.84 | 2.20 |
| HRFP-4 | 7 | 28.17 | 2.78 |
| | 28 | 28.87 | 2.83 |

In view of the values obtained, it can be concluded that adding fibres to the mixture, in the proportions tested, produces a slight increase in the compressive strength of the concrete. This improvement may be because the addition of fibres reduces the porosity of the concrete, obtaining a more compact and, therefore, more resistant material. The compressive strength of the three concretes after seven days is higher than the 16.5 MPa required for the CEM III used in the mix proportion.

To analyse the tensile strength, the values obtained at 28 days will be taken into account. The tensile strength of all the fibre-reinforced concretes was higher than the control value. In the case of the HRFP-2, the tensile strength increased by 6.28%, while for the HRFP-4 the increase was 36.71%. It can be concluded that there is a correlation between a greater volume of fibres and an increase in tensile strength.

In relation to the tensile strength of the cylinders, a change in the failure mode can be seen when the polyethylene fibres are added. Although the control specimens broke abruptly, typical of a brittle fracture in concrete, the specimens with fibres showed a smooth fracture, maintaining the original shape of the fibre. As can be seen in Figure 15, the fibres are responsible for holding the specimen in its position once the cracking occurs. It can be concluded that the incorporation of fibres decreases the brittle fracture of the concrete and also helps to control the cracks.

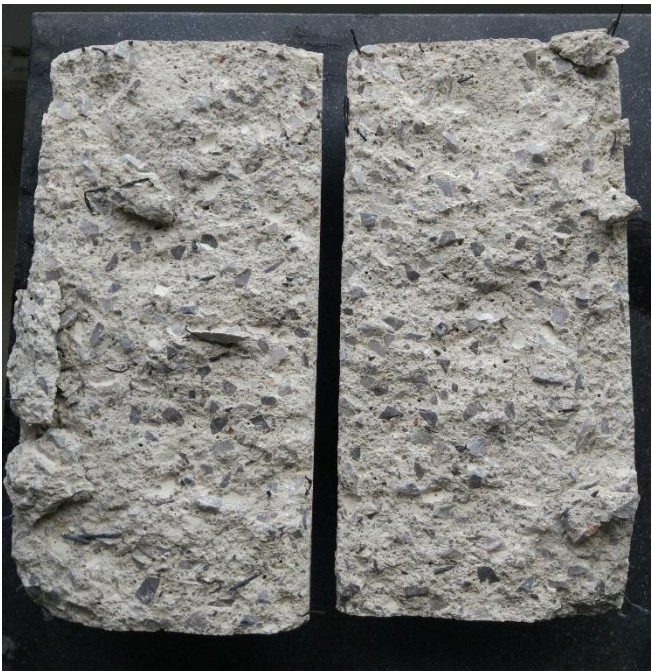

**Figure 15.** Breaking surface of a cylindrical specimen after a tensile splitting strength test.

Figure 16a,b shows a cut specimen, where the distribution of the fibres in the concrete can be seen; although the amount of fibres used is not very high, it can be seen that they are distributed randomly over the entire surface. This indicates that the polyethylene fibres mix easily with the concrete, without producing balls or hedgehogs. Furthermore, a good adhesion of the fibres can be seen, due to the roughness of the fibres, and other factors. It must also be taken into account that the direction of the fibres is random, which plays a fundamental role in the tensile strength of the concrete, as the fibres in the longitudinal axis to the specimen will not provide any reinforcement. Several authors have shown that the orientation and distribution of fibres play an important role in mechanical properties [68–71]. As an example, Figure 16c shows an example of the cylindrical specimen's fibre distribution.

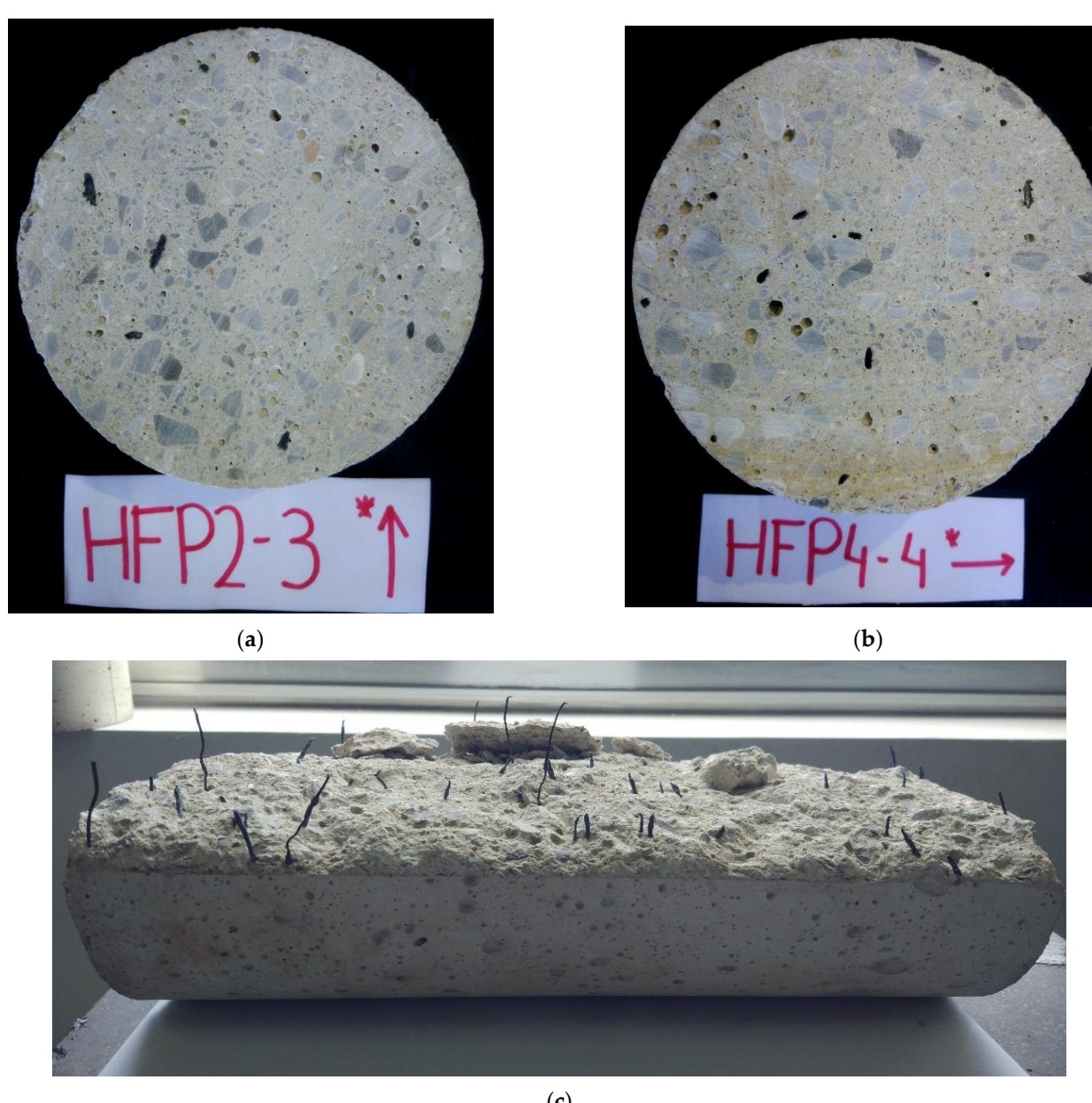

(**a**)　　　　　　　　　　　　　　　　　　　　　　(**b**)

(**c**)

**Figure 16.** Examples of fibre distribution (**a**,**b**) and the orientation of fibres (**c**).

### 3.4.4. Durability

Table 7 shows the results for concrete water and oxygen permeability of the three concretes.

**Table 7.** Concrete permeability results.

| Mix | Oxygen Permeability (m$^2$) | Water Permeability (mm) |
|---|---|---|
| HFC | $4.33 \times 10^{-18}$ | 59.85 |
| HRFP-2 | $6.32 \times 10^{-18}$ | 67.51 |
| HRFP-4 | $6.74 \times 10^{-18}$ | 52.41 |

Regarding the oxygen permeability, a comparison with the general concrete values shows that the three concretes manufactured have low permeability. On the other hand, the increase in the volume of fibres slightly increases the concrete's permeability to oxygen. This

may be because, although the fibres reduce the porosity of the concrete, the fibre material itself is permeable to oxygen, making it easier for oxygen to pass through the concrete.

As far as water permeability is concerned, EHE-08 requires a maximum water penetration of 50 mm. As the marks obtained are greater, it can be concluded that the concrete is quite permeable to water and, therefore, more exposed to potential damage. The high permeability shown is a consequence of a high water–cement ratio, established at 0.60.

## 4. Conclusions

In an effort to promote sustainability in construction, this work has focused on offering an application for industrial polyethylene waste, recycling it as fibres and using it as reinforcement in concrete. Later, an analysis of its mechanical properties was carried out. The following conclusions have been drawn after analysing the results obtained.

- The methodology followed for recycling the fibres does not adversely affect their mechanical behaviour; indeed, a material with better mechanical properties than the original one is obtained.
- The fibres obtained by injection have a very rough surface, which improves its adherence to the concrete.
- The addition of fibres to the concrete mix, in the proportions studied, produces a slight increase in the compressive strength of the concrete. This characteristic is linked to a 22% reduction in the porosity of the concrete.
- The tensile strength is one of the main advantages of the use of concrete reinforced with polyethylene fibres in comparison to the control concrete, since with $4 \, \text{kg/m}^3$ of fibres it is possible to increase resistance by 36.71%.
- Polyethylene fibres also demonstrate an excellent capacity for controlling cracks in concrete.
- A more thorough study of the durability of concrete with polyethylene fibres is required, as well as an analysis of the resistance of the fibres in an alkaline environment.
- Concretes with higher fibre percentages should be produced to check whether an increase in fibre content leads to an improvement in the tensile behaviour of the concrete.

The experimental results confirm the initial hypothesis concerning the potential of recycled polyethylene fibres for reinforced concrete structures.

**Author Contributions:** Conceptualization, C.T.; Data curation, J.A.S.-A. and M.S.; Formal analysis, J.A.S.-A.; Funding acquisition, C.T.; Investigation, M.S., L.G., P.T., G.G.d.A. and A.A.; Methodology, S.D., M.S., P.T. and C.T.; Project administration, C.T. and S.D.; Supervision, C.T.; Writing—original draft, J.A.S.-A.; Writing—review & editing, C.T. All authors have read and agreed to the published version of the manuscript.

**Funding:** This research was funded by the LADICIM (Laboratory of Materials Science and Engineering), Universidad de Cantabria. E.T.S. de Ingenieros de Caminos, Canales y Puertos, Av./Los Castros 44, 39005 Santander, Spain.

**Institutional Review Board Statement:** Not applicable.

**Informed Consent Statement:** Not applicable.

**Acknowledgments:** The authors would like to thank: LADICIM, the Laboratory of Materials Science and Engineering of the University of Cantabria, for making the facilities used in this research available to the authors. The authors would like to thank the "Augusto Gonzalez Linares" postdoctoral grant program of the University of Cantabria for their support. The authors would like to thank the "industrial doctoral" grant program of the Government of Cantabria for their support.

**Conflicts of Interest:** The authors declare no conflict of interest. The funders had no role in the design of the study; in the collection, analyses, or interpretation of data; in the writing of the manuscript, or in the decision to publish the results.

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
