# Peer review of "Recycled Polyethylene Fibres for Structural Concrete"

_applsci, doi:10.3390/app12062867_

Round 1
Reviewer 1 Report
The authors experimentally studied the possibility of adding recycled polyethylene fibers for concrete structures. The test data indicates that adding the recycled polymer fibers helps improve the mechanical performance of the concrete. As this study is of great practical importance in civil engineering, it is recommended for publication. However, there are a few issues that should be addressed:
- The statistical scatter of the fiber strength mentioned in line 294, could possible be captured by the Weibull’s weakest-link model (i.e. Weibull distribution).
- From line 361, what kind of “change in the failure mode” is observed? This can be hardly seen from Fig. 15. It is also unclear how the statement in line 365-366 is made that “fibers decreases the brittle fracture” and “helps to control the cracks”.
Author Response
Responses to the reviewers’ comments to the manuscript applsci-1618944 Recycled polyethylene fibres for structural concrete Dear editor, We greatly appreciate the opportunity you gave us to improve the paper with the valuable comments of the reviewers. Below you can find the detailed answers to the comments and resulting modifications. Best regards, The authors ---- Reviewer #1 (changes in blue) The authors done an extensive research on recycled polyethylene fibres for structural concrete, the below suggestions will help to improve the article Thank you very much for your efforts in reviewing this paper, we appreciate your help in improving this work. We will respond to all your comments below. 1. Include a brief details about valorisation Thanks for the suggestion. The following sentence has been added: “The exponential growth of waste makes it essential to look for recycling alternatives, such as valorisation. In addition, there is growing concern and studies associated with the valorisation of different materials [27–30].” 2. Add more information about Superplasticizer Thank you very much for the proposal, the following sentence was added to provide information about the superplasticiser: “MasterEase 3850 is a highly active superplasticising / water-reducing admixture based on new polymer technology for the production of low viscosity concrete even at low water contents.”. 3. For workability test which sand is used Thank you very much for the comment, by mistake, the types of sand were not defined: “The concrete mixture was made with siliceous sand 0-2 mm, limestone sand 0-3 mm and limestone gravel 6-12 mm.”. 4. Discuss more about Abrams cone test to check the workability Thank you very much for the suggestion. Based on your comment, the following text was added analysing the influence of fibres on the concrete workability: Based on the results shown in table 5, it can be seen that the presence of fibres barely changes the concrete workability. However, despite the limited influence of fibres on the fresh state behaviour, a slight reduction in the slump flow can be observed. This reduction in the workability of the concrete due to the presence of fibres agrees with the results obtained by other authors [67]. 5. fibres length, thickness, strength details can be incorporated in the manuscript Indeed, the properties of the fibres are the key to understanding their behaviour, so that Table 4 shows the recycled fibre properties, including length and diameter (thickness). Table 3 shows the mechanical properties of the recycled material. 6. For every test for studing the physical and other properties, the equipment used for studeny should be included Most of the tests carried out in this work are done in a standardised way, so the test tools are standardised and, from the authors' point of view, it is not necessary to indicate such a degree of detail. In any case, the most unusual elements have been identified and detailed, for example, on the basis of their commentary, the brand and model of the three-dimensional table have been added. Lines 150 to 151: “A three-dimensional table TESA - Micro-hite 3D was used to analyse the roughness of the fibres, as shown in Figure 4.”. Reviewer #2: (Changes in green) The manuscript entitled “ Recycled polyethylene fibers for structural concrete” refers to the actual problem of landfilling and polluting the environment with used plastic products. The authors showed the possibility of LDPE recycling and reusing as a reinforcement in concrete. Their tests, studies, results and analyses of polyethylene properties before and after recycling are quite promising. Thank you very much for your efforts in reviewing this paper, we appreciate your help in improving this work. We will respond to all your comments below. Moreover, the concrete properties are also good or better after the recycled polyethylene fibers addition. The only question is the cost of recycling polyethylene and producing fibers as a filler for concrete? Exact values are not available to the authors as the costs of valorisation processes depend to a large extent on the boundary conditions of the recycling. The manufacturing process of non-recycled granules consists of the acquisition of raw materials, generally a thermal process, extrusion and crushing. In the case of this study, the starting point is crushing, and the previous steps are not necessary. However, depending on the origin of the waste, different tasks may be necessary, such as logistical and cleaning costs. Logistical costs vary depending on where the waste is brought from, and can be very high. The cleaning tasks depend to a large extent on the previous origin of the waste, which can be very different. The proposed application would mean substantial savings, for example, in the case of direct application in a factory that manufactures this type of component and that recovers its own waste, clean and with hardly any logistical costs. Due to the above, it is not possible to make an economic analysis without specifying the case study. However, in any case, the current world situation makes it necessary to try to recover as much of the waste generated as possible. The manuscript is basically well written. I have one question about the sentence in Abstract: The aim of this comparison is to evaluate the process and analyse the reduction of the physical-mechanical properties in the process. What the authors mean? Thanks for the appreciation. The sentence has been completed in order to clarify it: “The aim of this comparison is to evaluate the recycling process and analyse the reduction of the physical-mechanical properties of the recycled polyethylene in the process.” In the introduction. The part from line 80-82, started from : "With this aim,...." is unnecessary. Thank you very much for your comment, it was deleted. Page 9, line 239, there should be written Shore Thank you very much for the correction, it was corrected in the manuscript. Conclusions, page 17, line409: ... 4kg/m3 needs correction. Thanks, it has been corrected: 4kg/m3. Page 10. lines 254-256 Subsequently, the average roughness of the fibers was analyzed, see Figure 11 (a), obtaining a value of 0.1173 mm with its corresponding standard deviation of 0.0352 mm, which indicates that the set of data obtained is homogeneous.. Thank you very much for the comment. The phrase "which indicates that the set of data obtained is homogeneous" was deleted, as the results are not homogeneous to a high degree. Reviewer #3: (Changes in RED) I have read the manuscript completely. The work is timely and interesting. I recommend it for publication after addressing my following comments. Thank you very much for your efforts in reviewing this paper, we appreciate your help in improving this work. We will respond to all your comments below. -The test process has been explained. However, there are still some questions. For example, how do you control the distribution of PR inside the concrete? Moreover, in which distribution pattern the PR can affect the concrete further? Definitely, in some distribution patterns of fibres, the mechanical and structural properties of recycled concrete will be enhanced more. Thus, an explanation is essential in this regard. Thank you for the suggestion, the following sentence has been added in section 3.4.3: “Several authors have shown that the orientation and distribution of fibres play an important role in mechanical properties [67–70]” -Authors mentioned that the size of granules is less than 5-6 mm. It means, there may be particles with size 1 mm or 6 mm, etc. From this reviewer’s point of view, using the same approximate size of PR particles may increase the mechanical properties of concrete further. The different sizes may result in more stress concentration due to the fact that stress contour moves from one point to another one along with the thickness of the composite. So, as far as the PR particles are not in the same dimensions, there would be stress concentration. In order to make very close sizes for the granular, authors could use multi-level sieves. So, a justification is required for this issue. After reading the reviewer's reply, we realised that the work had not been explained correctly in the paper, so some modifications were made to the text. This work shows the process of valorisation of polyethylene out of use as fibres for concrete. In order to manufacture these fibres, the first step is to manufacture polyethylene granules from end-of-life waste, these are the 5-6 mm particles. These particles are then fed into an injection moulding machine which produces fibres of approximately 60 mm in length and 1.73 mm in diameter. Only these recycled fibres were used in the concrete, not the particles. The following text was added to clarify this point. Line 81 to 82: In this context, this research involves the evaluation of out-of-use black polyethylene tubes, for their recycling and subsequent manufacture of reinforcing fibres which will be use for concrete. Reviewer #4: (Changes in GREY) The authors experimentally studied the possibility of adding recycled polyethylene fibers for concrete structures. The test data indicates that adding the recycled polymer fibers helps improve the mechanical performance of the concrete. As this study is of great practical importance in civil engineering, it is recommended for publication. However, there are a few issues that should be addressed: Thank you very much for your efforts in reviewing this paper, we appreciate your help in improving this work. We will respond to all your comments below. 1. The statistical scatter of the fiber strength mentioned in line 294, could possibly be captured by the Weibull’s weakest-link model (i.e. Weibull distribution). Indeed, it is very likely that the fibre property results follow a Weibull-type distribution. However, because only in the case of the fibres is there a sufficient number to fit the data to a statistical distribution (there are 5 standardised specimens of reference material, 10 of recycled material with standardised geometry and 40 of fibres), no such fit was made. 2. From line 361, what kind of “change in the failure mode” is observed? This can be hardly seen from Fig. 15. It is also unclear how the statement in line 365-366 is made that “fibers decreases the brittle fracture” and “helps to control the cracks”. In the case of concrete with fibres, when a tensile splitting strength test is performed, the specimen does not break into two pieces, since, after breaking the specimen, the fibres continue to sew the specimen together. However, this cannot be seen in fig 15, because in order to see the cross-section, the specimen has been forced to separate into two pieces. This is the reason why the brittle behaviour of the material is reduced, as it allows greater deformations before breaking and helps to control cracking.

Reviewer 2 Report
The authors done an extensive research on recycled polyethylene fibres for structural concrete, the below suggestions will help to improve the article
- Include a brief details about valorisation
- Add more information about Superplasticizer
- For workability test which sand is used
- Discuss more about Abrams cone test to check the workability
- fibres length, thickness, strength details can be incorporated in the manuscript
- For every test for studing the physical and other properties, the equipment used for studeny should be included
Author Response

(The authors gave the same response as above.)

Reviewer 3 Report
The manuscript entitled “ Recycled polyethylene fibers for structural concrete” refers to the actual problem of landfilling and polluting the environment with used plastic products. The authors showed the possibility of LDPE recycling and reusing as a reinforcement in concrete.Their tests, studies, results and analyses of polyethylene properties before and after recycling are quite promising.
Moreover, the concrete properties are also good or better after the recycled polyethylene fibers addition.
The only question is the cost of recycling polyethylene and producing fibers as a filler for concrete?
The manuscript is basically well written. I have one question about the sentence in Abstract:
The aim of this comparison is to evaluate the process and analyse the reduction of the physical-mechanical properties in the process.
What the authors mean?
In the introduction. The part from line 80-82, started from : "With this aim,...." is unnecessary. Page 9, line 239, there should be written Shore
Conclusions, page 17, line409: ... 4kg/m3 needs correction.
Page 10. lines 254-256 Subsequently, the average roughness of the fibers was analyzed, see Figure 11 (a), obtaining a value of 0.1173 mm with its corresponding standard deviation of 0.0352 mm, which indicates that the set of data obtained is homogeneous.
Author Response

(The authors gave the same response as above.)

Reviewer 4 Report
I have read the manuscript completely. The work is timely and interesting. I recommend it for publication after addressing my following comments,
-The test process has been explained. However, there are still some questions. For example, how do you control the distribution of PR inside the concrete? Moreover, in which distribution pattern the PR can affect the concrete further? Definitely, in some distribution patterns of fibres, the mechanical and structural properties of recycled concrete will be enhanced more. Thus, an explanation is essential in this regard.
-Authors mentioned that the size of granules is less than 5-6 mm. It means, there may be particles with size 1 mm or 6 mm, etc. From this reviewer’s point of view, using the same approximate size of PR particles may increase the mechanical properties of concrete further. The different sizes may result in more stress concentration due to the fact that stress contour moves from one point to another one along with the thickness of the composite. So, as far as the PR particles are not in the same dimensions, there would be stress concentration. In order to make very close sizes for the granular, authors could use multi-level sieves. So, a justification is required for this issue.
Author Response

(The authors gave the same response as above.)
